# Improving deep models of protein-coding potential with a Fourier-transform architecture and machine translation task

**Joseph D. Valencia**[1], **David A. Hendrix**[1,2]*

**1** School of Electrical Engineering and Computer Science, Oregon State University, Corvallis, Oregon, United States of America, **2** Department of Biochemistry and Biophysics, Oregon State University, Corvallis, Oregon, United States of America

* david.hendrix@oregonstate.edu

**Data Availability Statement:** Code for training and evaluating models and replicating figures is available at https://github.com/josephvalencia/

## Abstract

Ribosomes are information-processing macromolecular machines that integrate complex sequence patterns in messenger RNA (mRNA) transcripts to synthesize proteins. Studies of the sequence features that distinguish mRNAs from long noncoding RNAs (lncRNAs) may yield insight into the information that directs and regulates translation. Computational methods for calculating protein-coding potential are important for distinguishing mRNAs from lncRNAs during genome annotation, but most machine learning methods for this task rely on previously known rules to define features. Sequence-to-sequence (seq2seq) models, particularly ones using transformer networks, have proven capable of learning complex grammatical relationships between words to perform natural language translation. Seeking to leverage these advancements in the biological domain, we present a seq2seq formulation for predicting protein-coding potential with deep neural networks and demonstrate that simultaneously learning translation from RNA to protein improves classification performance relative to a classification-only training objective. Inspired by classical signal processing methods for gene discovery and Fourier-based image-processing neural networks, we introduce LocalFilterNet (LFNet). LFNet is a network architecture with an inductive bias for modeling the three-nucleotide periodicity apparent in coding sequences. We incorporate LFNet within an encoder-decoder framework to test whether the translation task improves the classification of transcripts and the interpretation of their sequence features. We use the resulting model to compute nucleotide-resolution importance scores, revealing sequence patterns that could assist the cellular machinery in distinguishing mRNAs and lncRNAs. Finally, we develop a novel approach for estimating mutation effects from Integrated Gradients, a backpropagation-based feature attribution, and characterize the difficulty of efficient approximations in this setting.

## Author summary

The ribosome is an information-processing cellular machine that synthesizes proteins from messenger RNA (mRNA) transcripts in a process called translation. Here, we model this information-processing behavior of the ribosome as an intelligent system capable of

bioseq2seq. Pretrained models and training data are available at https://osf.io/xaeqg/.

**Funding:** This work was supported by the National of General Medical Sciences (NIGMS) of the National Institutes of Health under award number R01GM145986 (DAV, JDV). JDV was additionally supported under the EECS Outstanding Scholar program at Oregon State University. The funders had no role in study design, data collection and analysis, decision to publish, or preparation of the manuscript.

**Competing interests:** The authors have declared that no competing interests exist.

predicting which RNA transcripts undergo translation and which do not. Modern artificial intelligence techniques can produce powerful models of biological sequence properties but sometimes fail to reflect the underlying biology despite high prediction accuracy. We show that training such a model to predict translated protein sequences from mRNA sequences improves its ability to distinguish mRNAs from long noncoding RNAs. This ability to predict translated RNAs also allows us to search for potential micropeptides, which are small translation products outside of conventional coding regions. We also introduce a neural network architecture to capture the 3-nucleotide periodicity that is known to occur in coding nucleic acids. Crucially, this novel setup helps to instill complex and biologically meaningful logic into the model, which we probe for clues as to the patterns that govern translation in living organisms.

## Introduction

The flow of genetic information from DNA to RNA to protein is a fundamental life process in which messenger RNAs (mRNAs) act as the information-carrying intermediaries. High-throughput sequencing has revealed the abundance of another class of RNA called long non-coding RNAs (lncRNAs), which share important biochemical features such as 5' capping and polyadenylation with protein-coding mRNAs [1]. LncRNAs are differentiated from smaller noncoding RNAs like tRNAs and microRNAs based on their greater length of at least 200 nucleotides (nt), and from mRNAs based on limited evidence of lncRNA protein expression and sequence conservation [2]. LncRNAs make up more than 68% of the human transcriptome and play important regulatory roles, particularly during development [3, 4]. They are implicated in numerous diseases including cancer and cardiovascular disease [5].

The protein-coding potential of many transcripts is unresolved, and many transcripts previously or currently annotated as lncRNAs are mislabeled and in fact possess small open reading frames (sORFs) that encode micropeptides [6]. Ribosome profiling (Ribo-Seq) shows that ribosomes bind readily to lncRNAs [7], though the ribosome does not interact with lncRNA ORFs in the same way as mRNAs, lacking a distinctive drop-off of Ribo-Seq coverage at ORF end [8]. Ribo-Seq protocols accounting for the 3-nt periodicity of ribosome footprint density [9] have identified some genuine sORF translation [10, 11]. Only a small fraction of the possible set of micropeptides encoded by transcripts currently annotated as lncRNAs have been directly detected via mass spectrometry, leaving the vast majority as presumptively nonfunctional or rapidly degraded [12–14]. Still, hundreds of putative lncRNAs have been confirmed to be misannotated, and these transcripts do encode micropeptides, for example, myoregulin, a 46-aa regulator of $Ca^{2+}$ activity that contributes to muscle cell performance [15]. Micropeptides are also involved in metabolism, red blood cell development, cardiomyocyte hypertrophy [16], inflammation, tumorigenesis and tumor suppression [17, 18], and more [19].

Such uncertainty as to the intrinsic protein-coding potential of ORFs raises the question of how cells distinguish true coding regions, with the translational machinery likely to play a critical role. Recent results suggest that general sequence features governing the kinetics of protein synthesis also separate mRNA and untranslated lncRNA ORFs more broadly [20]. The Kozak consensus sequence is well-characterized as the optimal context for translation initiation, and ribosomes can skip unfavorable AUGs through leaky scanning [21, 22]. Initiation can be affected by cis-regulatory features such as 5' UTR secondary structure [23] and upstream ORFs [24], and by trans-acting factors such as microRNAs [9] and RNA-binding proteins

[25]. Codon usage biases in the 5'-most region of the CDS are particularly known to affect the elongation rate during protein synthesis [26–28].

Distinguishing between mRNAs and lncRNAs is an important step in annotating newly sequenced genomes, and a variety of statistical and computational methods have been developed for this task. Codon Adaptive Index (CAI) [29] discriminates coding nucleic acids according to biases in the synonymous codons that code for each amino acid and Fickett scores [30] by the nucleotides present in the three codon positions. Early computational approaches used Fourier or wavelet analysis to identify coding sequences (CDS) from their characteristic periodicity of nucleotide identity induced by codon usage bias [31–34]. Machine learning methods have been designed around features such as the absolute length of ORFs, ORF length relative to the transcript, codon and hexamer frequencies including Coding Potential Assessment Tool (CPAT) [35] and coding potential calculator [36], and others [37, 38].

Although many prior machine learning methods achieve high classification performance, they typically rely on transcript-level summary features. Deep learning approaches can operate directly on sequences without such intermediate features and have proven effective in predicting properties of biological sequences, including a wide variety of functional genomics assays [39, 40], RNA splicing [41] and degradation [42], and protein structure [43]. A recent method called RNAsamba uses a convolutional neural network variant to achieve high performance from both nucleotide and amino acid sequence, but also relies on pre-defined features such as the longest ORF [44]. A critical limitation in the development of intelligent systems for classifying transcripts as protein-coding vs noncoding is the bias of using the translation and length of the longest ORF in machine learning approaches. Our group previously developed mRNN, the first recurrent neural network classifier of coding RNA from primary sequences alone [45]. There is a need for more flexible neural networks capable of learning sequence-specific rules that promote translation to better understand what drives translational efficiency. The advantage of these approaches is that they do not require feature engineering, and are capable of learning new biological rules that are encapsulated in the weights of the neural network. Interpretation of these deep neural networks can lead to the identification of new sequence features that are informative for the evaluation of biological sequences and understanding the regulation of translation. Interpreting deep models is challenging, but a significant literature in explainable artificial intelligence has arisen in regulatory genomics, with notable successes in uncovering transcription factor binding logic [46, 47]. Interpretation of similar deep models of protein coding potential could help identify new sequence features regulating translation.

In this paper, we describe bioseq2seq, a novel neural network model of biological translation based on the sequence-to-sequence (seq2seq) paradigm commonly used for machine translation of human languages. Although the genetic code follows a well-understood mapping between nucleic acid codons and amino acids, we demonstrate that learning to predict the protein sequence from the sequence of its message improves neural network performance in distinguishing mRNAs from lncRNAs. Adapting recent advances in token mixing neural architectures, we introduce Local Filter Network (LFNet), a computationally efficient network layer based on the short-time Fourier transform. We leverage perturbation-based feature importance values to extract sequence patterns which impact the model prediction and generate hypotheses about the regulatory elements that could differentiate coding RNA *in vivo*. Lastly, we offer evidence that while our LFNet-based bioseq2seq model robustly uncovers biological rules to learn protein-coding potential, it presents challenges for approximate interpretation techniques in deep learning. We address these challenges by introducing mutation-directed integrated gradients (MDIG), which we show has a strong correlation with synonymous sequence perturbations, and can be used to identify regions in transcripts that are important for defining protein-coding potential.

## Results

### Translation training objective improves classification performance

We downloaded lncRNA primary sequences and mRNAs matched with their encoded proteins and metadata from the NCBI RefSeq annotations of eight mammalian species. To explore the benefits of a translation-based learning objective, we used this dataset to construct several machine learning training tasks with varying levels of emphasis on predicting mRNA protein products. The first is a sequence-to-sequence task called bioseq2seq, in which we trained a model to output a class prediction of $\langle NC \rangle$ for lncRNAs or $\langle PC \rangle$ followed by a predicted protein sequence for coding mRNAs. We compared this to a baseline task called bioseq2class that only outputs the class prediction and not the protein sequence. As an intermediate subtype of bioseq2seq and bioseq2class, we investigated the impact of placing additional training weight on the leading classification token and early protein sequence positions relative to downstream tokens, which we refer to as bioseq2seq-weighted (bioseq2seq-wt). This task is parameterized by a weight $\lambda$ such that for $\lambda = 0$, bioseq2seq-wt approaches the standard bioseq2seq, for large $\lambda$ it approaches bioseq2class, and intermediate values effectively require models to learn a prefix of the protein translation for mRNAs. Finally, we trained a network called bioseq2start to predict the location of the start codon for mRNAs or the last index for lncRNAs, thereby implicitly learning where the ORF is but not the translation task. We used these methods to test the hypothesis that the translation task acts as a regularization that improves the performance in classification.

We designed a novel neural network layer, LFNet, to efficiently apply a short-time (local) Fourier transform to the high-dimensional vectors representing each input nucleotide and perform sequential updates via frequency-domain filtering. We compared LFNet layers to convolutional neural network (CNN) architectures for all training tasks. In our general encoder-decoder framework, we composed LFNet or CNN layers in an encoder stack to process the RNA. A stack of transformer decoders operates on the encoder hidden representations to produce an output, autoregressively consuming its own predictions to produce the next character, as necessary [49]. Within this general framework summarized in Fig 1, we optimized several architecture-specific hyperparameters, including hidden dimension and number of encoder and decoder layers, for every training task and encoder type separately for a fair evaluation (see Tables A and B in S1 Text). Every model ∈ {seq, seq-wt, class, start} × {LFNet, CNN} was tuned using the Bayesian Optimization Hyperband [50] algorithm and five replicates were trained at the best hyperparameter setting for each model.

We compared our models against prior methods: RNAsamba, a hybrid deep learning and feature-based classifier, as well as CPC2 and CPAT, two machine learning methods based on engineered features. These models were trained using our dataset and five replicates were also produced for RNAsamba. We report the classification performance on a withheld test set for our overall best and worst models alongside the prior methods in Table 1 and present a detailed breakdown of the F1 score for all eight model types in Fig 2A. At inference time, our bioseq2seq models produce a variable-length sequence, requiring heuristic search strategies such as beam search and normalization schemes to adjust for length bias. For this reason, decoding was halted after only the leading classification token was predicted. In this sense, although bioseq2seq was trained to predict full proteins, this capability does not directly influence the protein-coding probability at inference time. Our lowest-performing model according to average F1 across replicates was the LFNet architecture on the bioseq2class training task, ranging from 0.909 to 0.916 F1. The performance of this model is therefore roughly comparable to that of CPC2. The best model overall was LFNet on the bioseq2seq-wt task, achieving between 0.956 and 0.964 F1. This model setup clearly exceeds the performance of CPC2

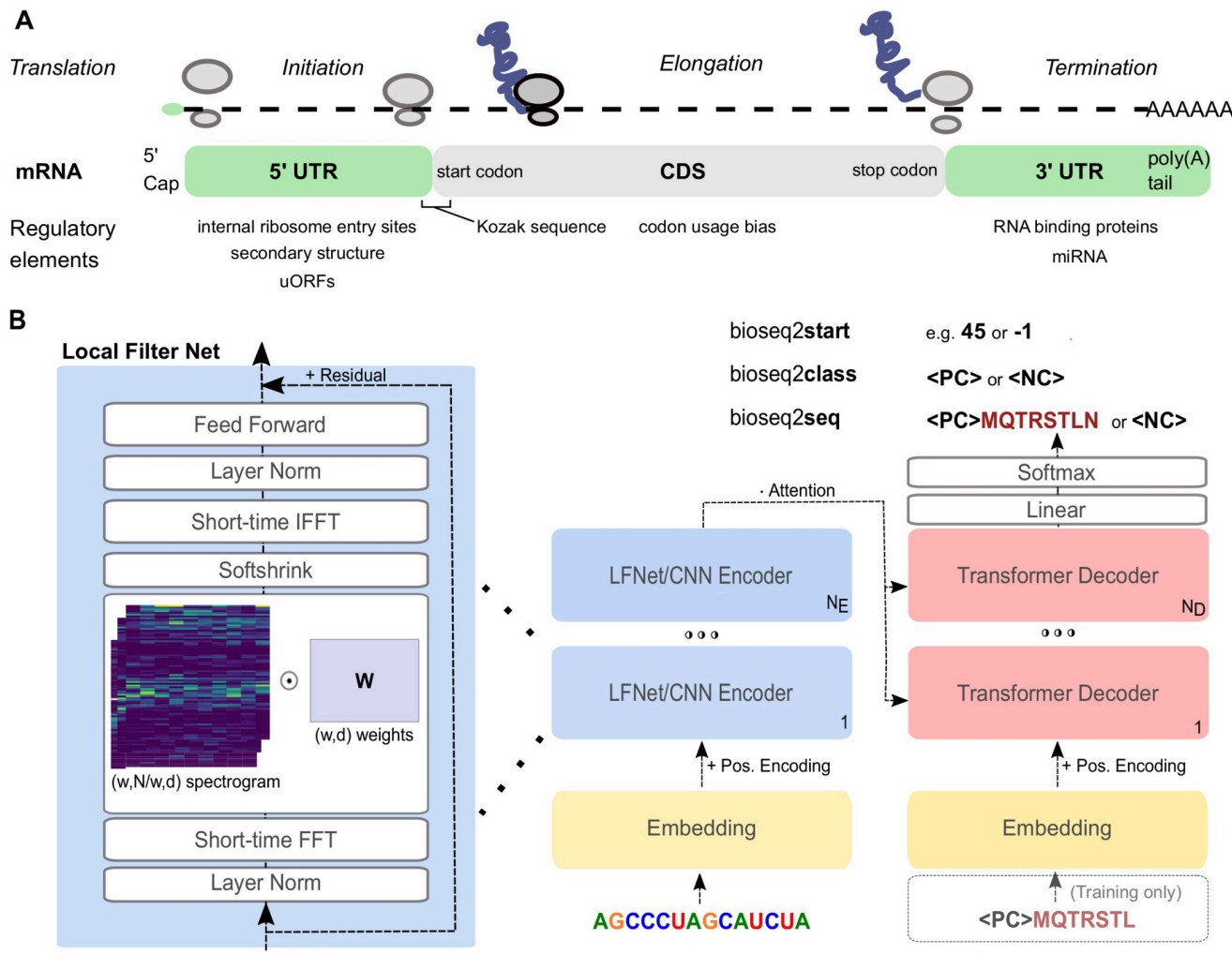

**Fig 1. Overview of problem setting and computational method.** (A) Summary of messenger RNA functional regions and known elements regulating translation. See [48] for a review of known regulatory elements. (B) Neural network sequence-to-sequence architecture. We designed LFNet (left) to apply a learned filter matrix $W$ to a 1D short-time Fourier transform (spectrogram) of the hidden representations, enabling frequency-domain filtering of the 3-base periodicity present in coding sequences. We trained this architecture for multiple problem settings: bioseq2class outputs a classification token, bioseq2seq also predicts the protein translation, and bioseq2start predicts the position of the start codon for mRNAs.

Table 1. Classification performance. Several versions of bioseq2seq, bioseq2start and bioseq2class models were trained on a dataset consisting of sequences from eight mammalian species. Our lowest and highest performing models are shown in this table alongside several top-performing machine learning models also trained on our dataset for comparison. For bioseq2seq-wt (LFN), predictions were made using the leading 'classification' token ⟨PC⟩ or ⟨NC⟩ of the first beam, terminating inference before the peptide prediction. For our models and RNAsamba, multiple replicates were trained with different random seeds. Evaluation metrics were calculated with ⟨PC⟩ as the positive class and listed as mean ± std. dev. where multiple replicates are available.

| Model | F1 | Recall | Precision | MCC | AUPRC |
|---|---|---|---|---|---|
| CPAT | 0.938 | 0.950 | 0.927 | 0.880 | 0.987 |
| CPC2 | 0.911 | 0.859 | 0.970 | 0.843 | 0.982 |
| RNAsamba | 0.956 ± 0.001 | 0.947 ± 0.002 | **0.964** ± 0.002 | 0.915 ± 0.002 | 0.992 ± 0.000 |
| bioseq2class (LFN) | 0.911 ± 0.003 | 0.896 ± 0.012 | 0.928 ± 0.015 | 0.832 ± 0.008 | 0.976 ± 0.002 |
| bioseq2seq-weighted (LFN) | **0.961** ± 0.003 | **0.963** ± 0.009 | 0.960 ± 0.012 | **0.925** ± 0.007 | **0.994** ± 0.000 |

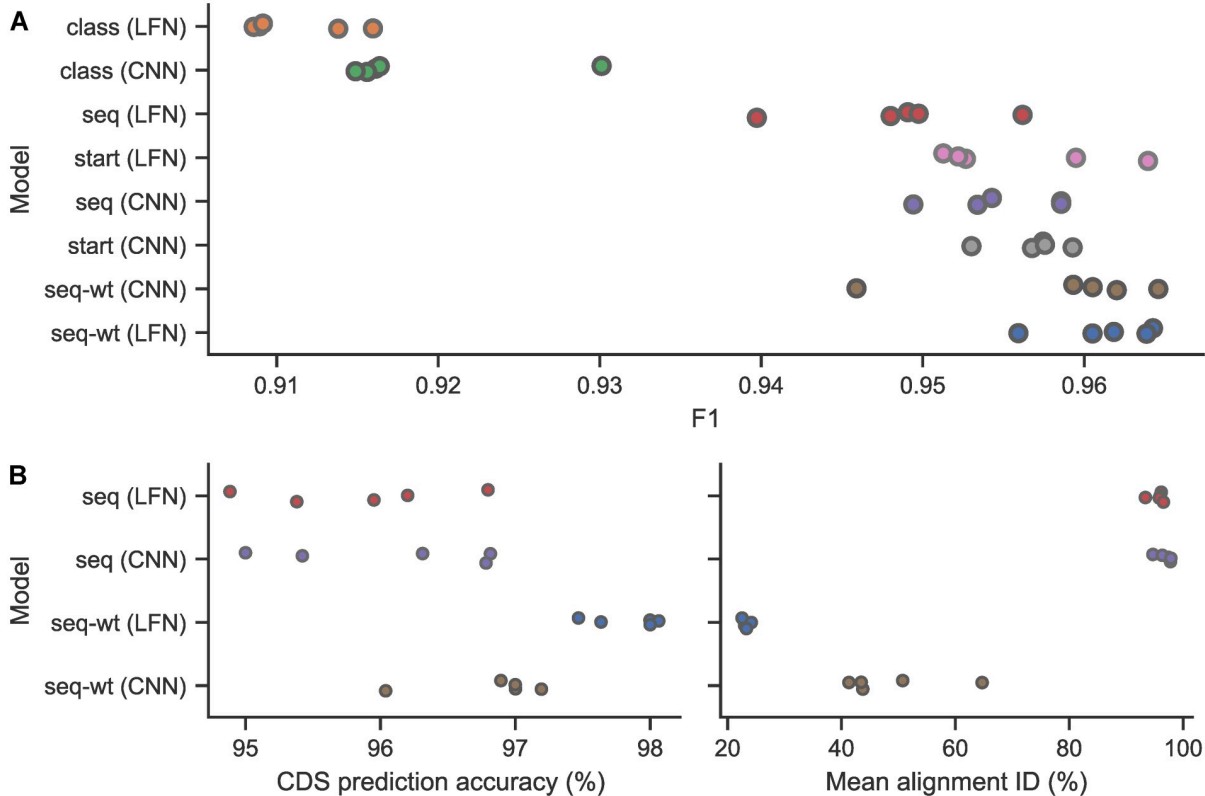

**Fig 2. Comparison of training tasks and neural architectures.** Names have been shortened by removing the "bioseq2" prefix for all of them. (A) F1 score across five replicates of bioseq2seq, bioseq2seq-wt, bioseq2class, and bioseq2start using both LFNet and CNN architectures. (B) Analysis of CDS detection abilities by bioseq2seq variants. Rate at which predicted protein sequence aligns better to the CDS than alternative ORFs (left), and alignment percent identity with the CDS (right).

and CPAT and largely surpasses RNAsamba (0.954–0.957 F1) without explicit inclusion of any auxiliary features such as ORF k-mers.

For the bioseq2class training task, the LFNet and CNN models score 0.911 and 0.919 on average F1 score, respectively. In contrast, every replicate for the translation-related tasks (bioseq2seq, bioseq2seq-wt and bioseq2start) exceeds 0.94 F1. This substantial gap makes it clear that injecting information on the protein translation into the learning process improves the performance of deep models on the binary classification task. There was also variation between the translation-related tasks—in general, the bioseq2seq-wt objective outperforms bioseq2-start, which outperforms unweighted bioseq2seq. See the Methods section for bioseq2start implementation details. The CNN architecture tends to outperform LFNet on the same task, with the one exception being bioseq2seq-wt. The LFNet architecture and bioseq2seq-wt task is the best overall training setup according to its mean F1 of 0.961 against 0.958 for the CNN version, though the best single CNN replicate is marginally higher than that of LFNet (0.9646 vs 0.9643). Extended classification results are presented in the Supplementary Methods (S1 Text), including model precision-recall curves (Fig A in S1 Text), comprehensive metrics for both our models and the prior work (Table C in S1 Text), and an analysis of performance according to homology between the testing and training sequences (Fig B in S1 Text). The classification performance of bioseq2seq-wt suggests that learning to translate prefixes of the encoded proteins is the optimal training strategy, outperforming the direct classification task, start codon prediction, and full protein prediction.

Although the translation-related tasks were designed to improve model predictions of protein-coding potential, they predict the sequence or location of expressed proteins as a byproduct. To explore the accuracy of these predictions, we first extracted and translated all sense ORFs from every mRNA sequence in the test set. Unlike in the previous section, where decoding was halted after the initial classification token, we ran full beam-search decoding with the bioseq2seq and bioseqseq-wt models to predict full-length protein translations. Using bioseq2seq models in this fashion can alter the classification performance as beam search requires adjusting raw probabilities to account for the variable length of hypotheses. For instance, the best bioseq2seq-wt LFNet model deteriorated slightly to 0.963 F1. Focusing only on the true positive mRNAs for the bioseq2seq variants, we used the EMBOSS package to find the top Needleman-Wunsch alignment identity between the the top-beam protein prediction against each hypothetical translation product. In Fig 2B, we report the rate at which the annotated protein has the highest alignment percent identity with the bioseq2seq prediction, along with the mean sequence identity between the bioseq2seq prediction and the ground truth protein across transcripts. Notably, the details of translation fidelity have little to do with the capability to detect the correct CDS, with bioseq2seq-wt achieving over 98% accuracy in CDS detection despite alignment identities around 25%. Accuracy at the amino acid level is driven mainly by the $\lambda$ parameter and bioseq2seq-wt (LFN) optimized classification performance at a value of $\lambda = 0.1$, higher than the $\lambda = 0.05$ for bioseq2seq-wt (CNN). The bioseq2seq-wt models accurately align to a prefix of the correct translation, achieving slightly higher translation accuracy in the earliest protein characters than other methods (see Fig C in S1 Text), which could explain why they are the most adept at identifying the proper CDS location despite untrustworthy predictions on full-length sequences. Such details on translation performance provide additional context for how the auxiliary translation objective is related to protein coding potential.

## Alternate decodings of lncRNAs harbor plausible micropeptides

The bioseq2seq formulation can produce and rank multiple candidate decodings for a given RNA using beam search. For sequences annotated as lncRNAs and correctly classified by bioseq2seq, the lower beams (second highest scoring and on) will with high probability begin with $\langle PC \rangle$. We investigated the predicted peptides for insights into the potential translation of lncRNAs. First, we confirmed that the peptides predicted by unweighted bioseq2seq matched a true ORF within the lncRNA by computing global alignments between the three-frame translation and bioseq2seq-generated peptides from a beam size of four [51]. In 56.6% of cases for the best CNN replicate and 49.3% for the LFNet, the best match was a perfect alignment, meaning that alternate predictions for lncRNAs generally include translations of ORFs actually present in the lncRNA.

We applied bioseq2seq to a set of transcripts previously or currently annotated as lncRNAs but considered by the database LncPEP to have been validated by supporting literature to express a micropeptide (Table 2) [52]. Starting from the LncPEP "validated" set, we implemented a number of quality control measures, removing redundant transcripts, linking the transcript names listed on the LncPEP website with RefSeq accession numbers via the underlying primary literature and the NCBI search function. This yielded twenty-two putative micropeptide-encoding transcripts (provided as S1 Table). We built a modified training set by removing the LncPEP transcripts and their homologs, then retrained one replicate for each of the translation-related models on this dataset.

We ran the bioseq2seq variants in full-protein prediction mode and counted the number of LncPEP sequences predicted to be protein-coding and the subset of those for which the annotated micropeptide was the best match for the predicted peptide. Similarly, we evaluated the

**Table 2. Results on twenty-two validated micropeptides, separated into transcripts where the longest ORF corresponds to the true CDS ("Longest ORF", n = 10), and those where it does not ("Not Longest ORF", n = 12).** Each model was evaluated on classification accuracy (percentage predicted to be coding) and the percentage of the CDSs that were identified.

| Model | Longest ORF (n = 10) | | Not Longest ORF (n = 12) | |
|---|---|---|---|---|
| | Pred. Coding | Pred. CDS | Pred. Coding | Pred CDS |
| bioseq2seq (LFN) | 7 (70.0%) | 7 (70.0%) | 6 (50.0%) | 2 (16.7%) |
| bioseq2start (LFN) | 4 (40.0%) | 4 (40.0%) | 6 (50.0%) | 3 (25.0%) |
| bioseq2seq-wt (LFN) | 6 (60.0%) | 6 (60.0%) | 3 (25.0%) | 1 (8.3%) |
| bioseq2seq (CNN) | 6 (60.0%) | 6 (60.0%) | 4 (33.3%) | 2 (16.7%) |
| bioseq2start (CNN) | 6 (60.0%) | 6 (60.0%) | 4 (33.3%) | 3 (25.0%) |
| bioseq2seq-wt (CNN) | 2 (20.0%) | 2 (20.0%) | 2 (16.7%) | 0 (0.0%) |
| RNAsamba | 4 (40.0%) | 4 (0.0%) | 1 (8.3%) | 0 (0.0%) |

bioseq2start models and an RNAsamba model trained on the same reduced dataset. The unweighted bioseq2seq models performed best, with the LFNet version as the best overall. We see that for the transcripts described in Table 2, the micropeptide is not always coded for by the longest ORF. The bioseq2seq models are also the top performing on these challenging examples, demonstrating that they are capable of considering cryptic sources of coding potential. RNAsamba only explicitly considers the longest ORF in each transcript and deterministically outputs its translation for predicted coding sequences. Thus it may fail to identify alternate sources of coding potential, as here it predicts only one such example as coding and inherently cannot localize its translation. The translation product for AW112010.1 comes from an instance of non-AUG initiation [53], and while bioseq2seq cannot perfectly predict the protein product in such cases we successfully identify it as a coding transcript and predict a partial match from the canonical portion of the CDS.

## Local Filter Networks emphasize 3-nt periodicity

The core feature of each LFNet layer is its learned frequency-domain filters. We visualized the filter weight matrices to investigate the frequency response of the model to signals in the intermediate vector representations, including separate plots for their magnitude $|z|$ and phase $\theta$ for the complex weights $z = |z|e^{i\theta}$. The resulting images for all layers in both bioseq2seq-wt and bioseq2class are given in Fig 3. Visually, the most prominent signal in both model types is a band at a frequency bin equivalent to a period of 3 nt. This illustrates that most layers and hidden dimension across the LFNet stack learned to emphasize the 3-base periodicity of coding regions. Notably, every layer of bioseq2class (panel B) shows a more clear dependence on the 3-nt property than bioseq2seq-wt (panel A), with every layer having a clean visual band of low magnitudes along this period range. In contrast, lower layers of bioseq2seq-wt do not appear to emphasize this feature. However, bioseq2seq has phase values close to zero along the 3nt band (panel C), while the phase activity of bioseq2class is somewhat more random (panel D). We observed in Fig D in S1 Text that for bioseq2seq-wt, the periods other than 3-nt are associated with phases peaked around $-\pi$ and $\pi$, which correspond to phase components of the weights being $e^{i\theta} = -1$, such that the output of the LFNet layer would negate the residual. We hypothesize that the inductive bias of LFNet facilitates a reliance on the 3-base property, and the translation task leads to the amplification of specific 3-base signals.

Three-base periodicity is also apparent in our models' encoder-decoder attention (EDA) distributions, which are probability weightings for encoder hidden embeddings in the context of each decoder layer. We aligned each encoder-decoder attention distribution for every

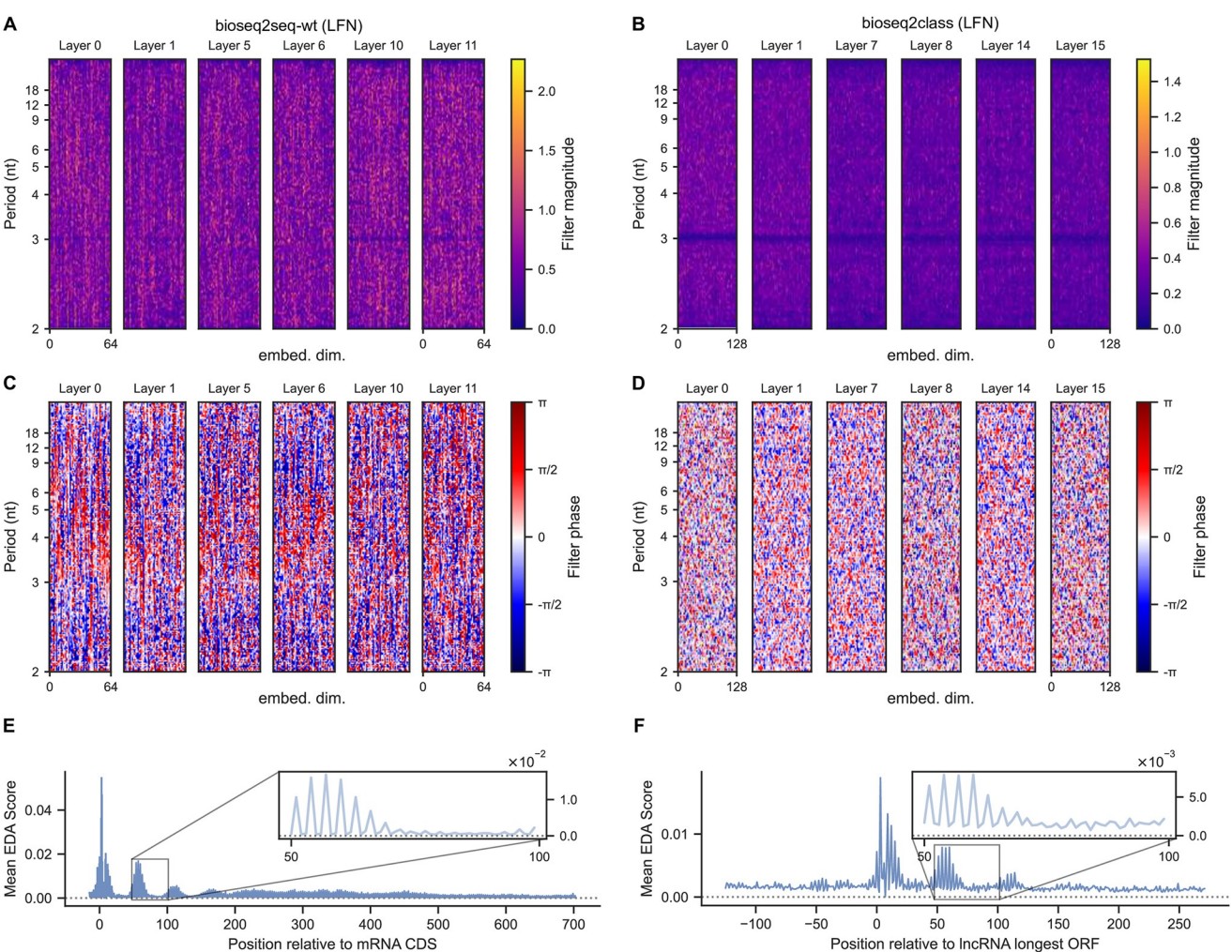

**Fig 3. Frequency-domain content in model representations.** LFNet filters from selected layers, with complex filter weights visualized in terms of magnitude (bioseq2seq-wt in panel A, bioseq2class in B) and phase (bioseq2seq-wt in C, bioseq2class in D). For each layer heatmap, the x-axis represents the hidden embedding dimension, and the y-axis refers to a discrete frequency bin, with annotations for the equivalent nucleotide periodicity. Both model types learned weights with a pronounced structure around 3-nt periodicity, visible mostly clearly in the phase for bioseq2seq-wt and in the magnitude for bioseq2class. (E) A nucleotide-resolution metagene consisting of average encoder-decoder attention scores from mRNAs aligned relative to their start codons. Attention distributions for this plot were taken from head 5 of the lower bioseq2seq-wt (LFN) decoder layer, which primarily attends to the start codon and places attention downstream of the start in a periodic fashion. (F) The equivalent plot for the same attention head applied to lncRNAs aligned relative to the start of the longest ORF.

transcript relative to its start codon and averaged to create nucleotide-resolution consensus attention metagene plots. For lncRNAs, we investigated the longest ORF to define metagenes and to compare and contrast mRNAs and lncRNAs in the rest of this manuscript. We considered the two classes separately and discarded relative positions not present in at least 70% of the data, leaving relative position indices of (-15,+703) for mRNAs and (-125,+271) for lncRNAs. Depicted in Fig 3 are metagenes for a particular EDA head in the lower decoder layer of bioseq2seq-wt (LFN) computed from mRNAs (panel E) and lncRNAs (panel F), attending highly to the AUG/longest ORF in both classes, with subtle differences in attention behavior that likely implement aspects of the model's classification logic. We present more detailed analysis of EDA metagenes in Fig E in S1 Text.

## Translation task improves reproducibility and biological plausibility of variant effect predictions

We evaluated all replicates from our encoder-decoder based models on every possible single-nucleotide variant of transcripts from a subset of our test data, consisting of 146 verified mRNAs and 140 verified lncRNAs. This technique, known as saturated *in silico mutagenesis* (ISM), is commonly used to computationally predict variant effects and can provide insight into input features that machine learning models recognize as important to their predictive task [40, 54, 55]. We calculated ISM using the function

$$\Delta S(x, x') = \log(\frac{P(x' = \langle PC \rangle)}{P(x' = \langle NC \rangle)}) - \log(\frac{P(x = \langle PC \rangle)}{P(x = \langle NC \rangle)})$$

where $x$ and $x'$ are RNAs, with $x'$ being a single-nucleotide variant of $x$. We calculated the Pearson correlation between the ISM scores predicted by two different replicates for a given transcript, making pairwise comparisons between all replicates. These correlations were averaged across comparisons to produce a single value for each transcript, with the resulting distributions depicted in Fig 4B. The inter-replicate agreement is much higher for all of the bioseq2seq variants, particularly the weighted models (LFN median $r = 0.857$, CNN median $r = 0.901$), than for bioseqclass (LFN median of $r = 0.526$, CNN median of $r = 0.311$). We also computed the cosine similarity between the character-level (A,C,G,U) vectors of mutation scores at each transcript position, using the median of this quantity as a transcript-level summary metric that does not compare the scaling of mutation scores at different positions. Inter-replicate comparison with this relaxed metric shows a minimal difference between the LFNet version of bioseq2class (median of 0.592) and the bioseq2seq-wt (LFN median of 0.566, CNN median of 0.579) models. This suggests that the gap in reproducibility between the model types is largely due to bioseq2seq's more stable ranking of positional importance.

We next evaluated model effect predictions on sequence perturbations disrupting essential mRNA features. First, we shuffled every 5' UTR longer than 25 nt in the verified test set, using a shuffle preserving dinucleotide frequencies, and likewise for 3' UTRs separately. We calculated $\Delta S$ for each shuffled variant relative to its wild-type and found that UTR shuffling had minimal impact on on the predictions of either bioseq2seq-wt or bioseq2class (Fig 4C). Similarly, we took the 5' UTRs for mRNAs that did not contain an upstream ORF and swapped them for 5' UTRs from uORF-containing mRNAs, finding that $\Delta S$ was slightly negative on average (Fig F in S1 Text). We produced another set of variants by shuffling all codons besides the start and stop codon within CDS regions. This has the effect of preserving the original CDS length while likely disrupting 3-nt periodicity and leading to atypical orderings of nucleotides and amino acids. Shuffling codons was much more strongly deleterious to coding potential than shuffling UTRs. Bioseq2class appears somewhat more reliant on the endogenous trinucleotide patterns of wildtype CDS regions than bioseq2seq, as indicated by the stronger negative $\Delta S$ after shuffling internal codons of CDS sequences. In contrast, mutations to the annotated start codon tended to produce large negative $\Delta S$ scores in bioseq2seq-wt but not in bioseq2class (Fig 4D). Mutations that introduced a stop within the first 50 codons received a highly negative $\Delta S$ score from the bioseq2seq-wt models. Stop codons introduced in bins further downstream continued to be scored negatively by the bioseq2seq-wt models, with the effect size steadily lessening across the length of transcripts. In contrast, bioseq2class learned relatively little about nonsense mutations. These observations suggest that while both models detect periodic sequence features, bioseq2seq-wt has learned contextual sequence features, including start and stop codons, that more comprehensively align with our understanding of translation.

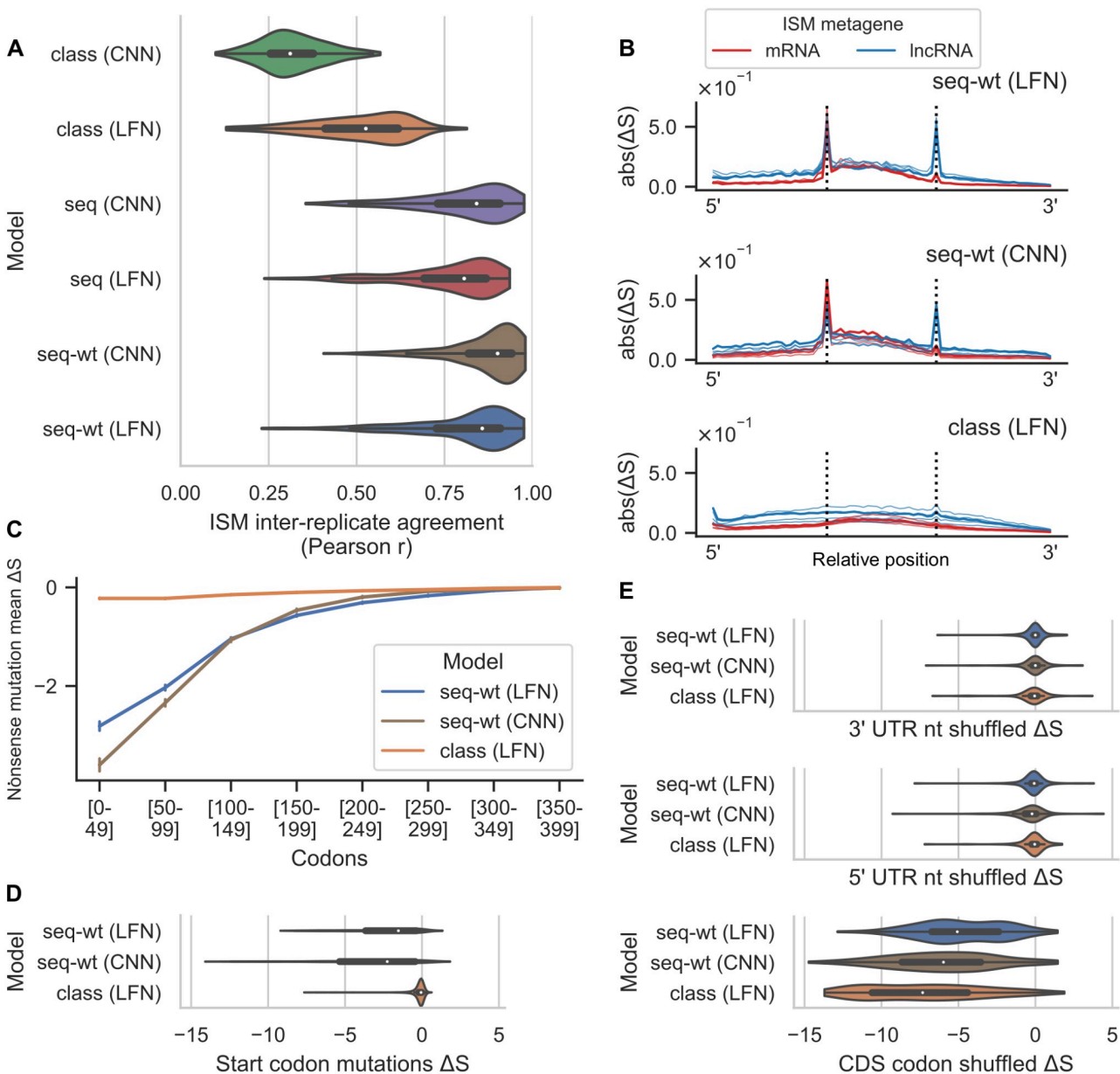

**Fig 4. Predicted mutation effects by model type on a subset of testing data.** Names have been shortened by removing the "bioseq2" prefix for all of them. (A) Inter-replicate agreement according to Pearson correlation of saturated in silico mutagenesis (ISM) $\Delta S$ scores, i.e. the difference in $\log(P(\langle PC \rangle)/P(\langle NC \rangle))$ between single-nucleotide variants and their wild-type sequence. Correlation of ISM scores is computed pairwise across replicates and averaged into a single value per transcript. (B) Metagene plots of ISM in which the absolute value of $\Delta S$ was averaged within each of 25 positional bins and across all three possible mutations in each position, with mRNAs and lncRNAs depicted separately for both bioseq2seq-wt (LFN), bioseq2seq-wt (CNN), and bioseq2class (LFN). Vertical dashed lines denote the first and last bin of the CDS for mRNAs and the longest ORF for lncRNAs. Metagenes from all five replicates are shown, with the best-performing model colored using the darkest hue. (C) Changes in coding score for changes that introduce a premature stop codon, in fifty-codon bins along the length of the CDS. (D) Changes in score for mRNAs from nucleotide substitutions that knock out a start codon. (E) Changes in score relative to wildtype for mRNAs shuffled within each functional region. UTRs were shuffled to preserve dinucleotide frequencies. Codon shuffling excluded the start and stop codons to preserve CDS length.

## In silico mutagenesis reveals features predictive of coding potential

In light of the improved biological robustness demonstrated by models trained with a translation objective, we proceeded to further interpretation of the LFNet version of bioseq2seq-wt. Using its best replicate, we obtained ISM predictions for the remainder of the test set. We aggregated ISM scores for all synonymous point mutations inside of mRNA CDS regions into fine-grained metagene plots for each amino acid, computing the mean $\Delta S$ along each of 25 positional bins. Selected amino acids are highlighted in Fig 5A and all twenty are depicted in Fig G in S1 Text. As expected for a highly contextual model, there are large deviations away from the mean. On average however, the amino acids with only two codons all learn a preference for a single codon across the length of the whole transcript, with correspondingly negative scores for the opposite mutation. The amino acids with more than two-fold degeneracy are more complex to interpret but the sign for the mean mutation effect tends not to change with position. When considering all synonymous mutations, the model appears to have learned a preference for particular nucleotides in the codon positions. For example, most codons ending in C have a positive $\Delta S$ on average, and most ending in U have a negative effect (Fig 5B). Bioseq2seq's estimates of synonymous mutation effects also captured some of the variation from an external measure of translation efficiency called tRNA Adaptation Index (tAI) [56] (see Fig H in S1 Text). The mean $\Delta S$ for point mutations leading to synonymous changes show a moderate correlation ($r = 0.449$, $\rho = 0.488$) with the differences in tAI between the two codons, using codon values calculated from [26].

We used ISM scores as a feature explanation method by assigning each nucleotide within a transcript an importance score based on the magnitude of $\Delta S$ from the mutation in that position that most disrupts bioseq2seq classification towards the opposite class. For example, an endogenous $x_i$ within an mRNA was defined as contributing towards a true positive classification of the $\langle PC \rangle$ class to the extent that substituting any of the three alternate bases in position $i$ produced a highly negative $\Delta S$.

One representative example mRNA and lncRNA are visualized in Fig 5C and 5D, respectively, with raw ISM scores from positions of interest shown in a heatmap. The transcript sequences are overlaid above with their heights drawn proportionally to the importance setting for their true class—↑ PC for the mRNA and ↑ NC for the lncRNA. The samples were chosen from among the five lncRNAs and mRNAs closest to the median value for inter-replicate agreement (see Fig 4B). In the example mRNA, the start codon is a highly salient region, while the stop codon receives little importance. The ISM scores for the nucleotides surrounding the start codon imply a preference for G in position + 1 relative to the start, consistent with the Kozak consensus sequence. The most important feature occurs where multiple mutations would introduce a stop codon less than fifty codons into the CDS, confirming our observation that the model disfavors nonsense mutations early in the coding sequence. For the lncRNA, the TGA ending the longest ORF receives high importance according to ↑ NC, but a TAA upstream of the longest ORF is the highest overall.

To systematically extract general patterns that bioseq2seq recognizes as predictive of coding potential, we performed de novo discovery of motifs frequently found in transcript subsequences with high ISM importance. First, we identified the most important nucleotide with respect to both ↑ PC and ↑ NC from each functional region (5' UTR, CDS, 3' UTR) of each test-set mRNA and likewise for the regions demarcated by the longest ORF of a lncRNA. We extracted 21-nt windows centered around each such important site to form a primary sequence database for the differential motif discovery tool STREME [57]. A control set for STREME was constructed either using (1) random positions from the same transcript and region as the primary sequences but not overlapping them (2) the most important positions

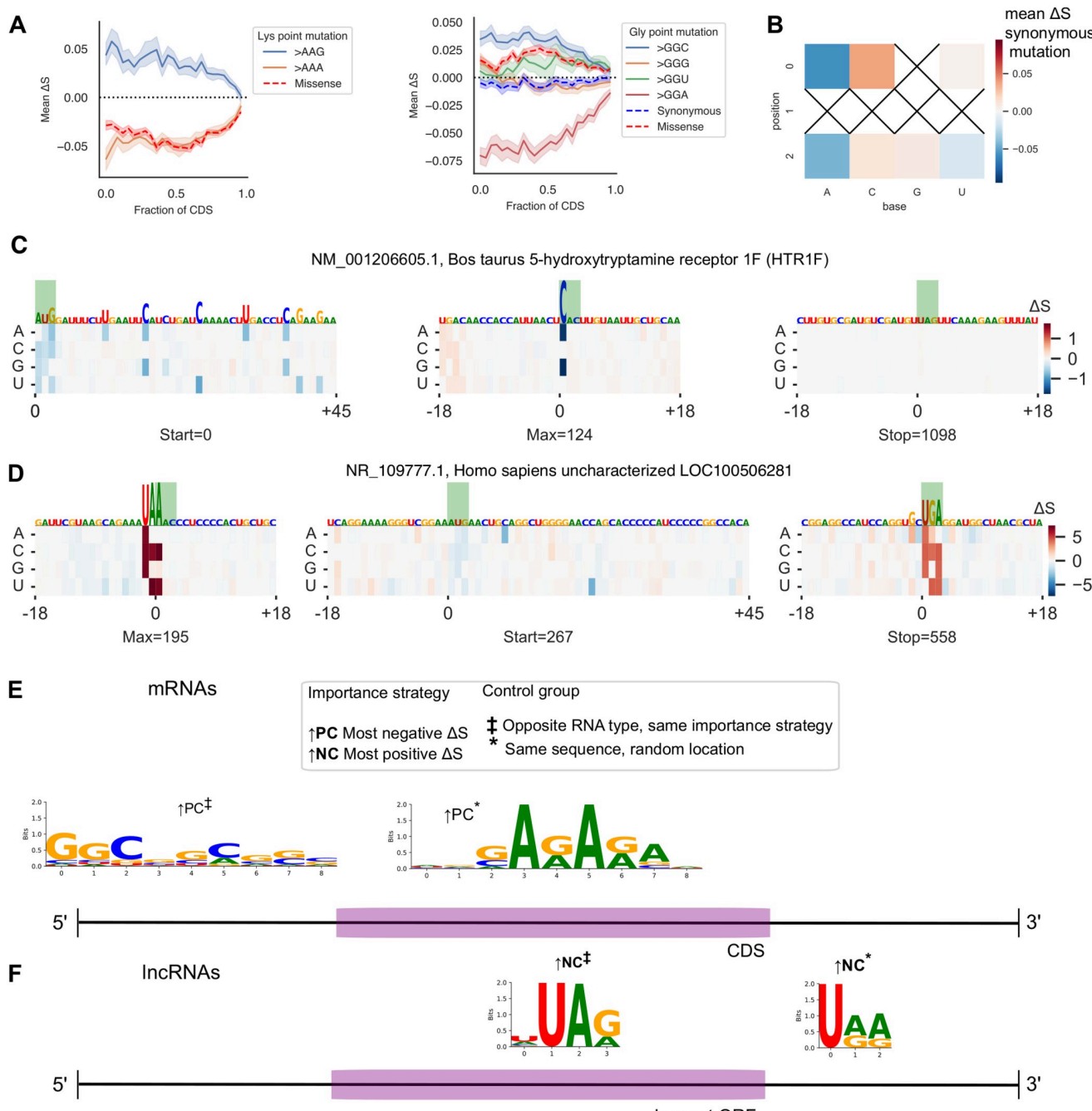

**Fig 5. Detailed analysis of *in silico* mutagenesis (ISM) on the full test set.** (A) Plots of ISM metagenes for selected amino acids lysine (left) and glycine (right). Mean $\Delta S$ is shown for 25 positional bins across mRNA CDS regions with mutations listed based on the resulting codon. The red line represents the average across all missense/nonsynonymous mutations. For amino acids with more than two codons, the blue dashed line depicts the average synonymous mutation for comparison. (B) Mean ISM for synonymous point mutations by codon position and nucleotide. X's denote substitutions which do not exist as synonymous changes. (C) An example protein-coding transcript with NCBI accession NM_001206605.1. Signed ISM scores for the transcript are depicted as a heatmap and the RNA sequence is portrayed with characters scaled according to the $\uparrow PC$ importance strategy, i.e. regions with highly negative ISM weights depicted in dark blue. The subregions shown are windows around the start codon, the position of maximum importance, and the stop codon. (D) Same as panel C with an example long noncoding RNA with NCBI accession NR_109777.1. The endogenous sequence is scaled according to $\uparrow NC$, or highly positive ISM values drawn in dark red. (E) mRNA motifs discovered in our test set with STREME using ISM importance values from bioseq2seq to determine sequence regions in which to search for enriched signals. Annotations denote the importance and control strategy for each trial, with boldfaced annotations signifying that importance values were not masked and ordinary typeface indicating that feature importance at start and stop codons and nonsense mutations were excluded. Motifs are positioned near the regions in which they were enriched. (F) Same as panel E showing discovered lncRNA motifs.

using the same importance setting as in the primary sequence but from the opposite RNA class. These controls necessitate different interpretations of the discovered motifs, with strategy 1 intended to establish whether bioseq2seq places importance on consistent features of a transcript, and strategy 2 intended to uncover differences in how bioseq2seq treats roughly comparable regions of coding and noncoding transcripts. We also ran motif discovery using a purely random strategy—e.g. with randomly chosen subsequences of a 5' UTR as primary and random upstream regions of a lncRNA as control. We present only strategy 2 motifs that do no match a motif from the purely random trials according to TOMTOM [58], as these experiments were specifically guided by ISM importance.

We ran every combination of primary sequence region, control method, and importance setting as its own STREME experiment and discovered two significant motifs in lncRNAs. Finally, we ran a second set of experiments in the same manner except with importance scores for endogenous start and stop codons and counterfactual missense mutations masked out in order to reveal important signals beyond the most prominent set found in the first run. This yielded two additional motifs, and both sets are shown in Fig 5E for mRNAs and F for lncRNAs, with boldface annotations for the unmasked motifs. The experiments with random controls largely confirm the observations we made in our example transcripts, with several stop codon trinucleotide motifs prominent throughout lncRNAs. Repeated GA patterns appear enriched in important CDS regions that push mRNAs towards a true positive classification, while GC rich elements occur more often in the 5' UTRs. Additional details including positional and frame biases and enrichment, can be found in S2 and S3 Tables.

## Approximation quality of gradient-based mutagenesis depends on model complexity

Saturated ISM is costly to apply to a large amount of sequences because it requires $3L$ model evaluations, where $L$ is the transcript length. We explored the feasibility of approximating ISM using neural network input gradients, which are efficiently computable in parallel via automatic differentiation. Building from the Integrated Gradients (IG) method, we developed a novel proxy for ISM called Mutation-Directed Integrated Gradients (MDIG). MDIG involves numerically integrating input-output gradients along the linear interpolation path between a sequence of interest and a sequence of the same length consisting of all the same type of nucleotide, e.g. all guanines. A parameter $\beta \in (0, 1]$ limits how far to travel towards the $poly(b)$ baseline embedding during integration. (See Methods). As a favorable value for $\beta$ is not obvious from first principles, we tuned this parameter on a subset of our validation set consisting of 206 verified mRNAs and 206 lncRNAs.

We then computed the per-transcript Pearson correlation of scores from different settings of MDIG-$\beta$ with the ISM scores from the same replicate. This metric indicates MDIG's capacity to approximate the input-output behavior of a given deep learning model, which ISM accomplishes directly but at substantially greater computational cost. For reference with other gradient-based perturbations, we perform the same analyses using a first-order Taylor approximation of ISM scores and IG with a uniform [0.25, 0.25, 0.25, 0.25] baseline over the four nucleotides. Results on the validation set according to this evaluation metric are summarized with their median value in Fig 6A. On the basis of these results, MDIG-0.25 was selected as the best approximation method for bioseq2seq-wt and MDIG-0.1 for bioseq2class. This illustrates that the MDIG method can predict the effect of input perturbations better than the basic Taylor approximation.

On the whole, we observed a large gap in approximation quality between the model types, with the best method for bioseq2seq-wt lagging substantially behind the worst for bioseq2class.

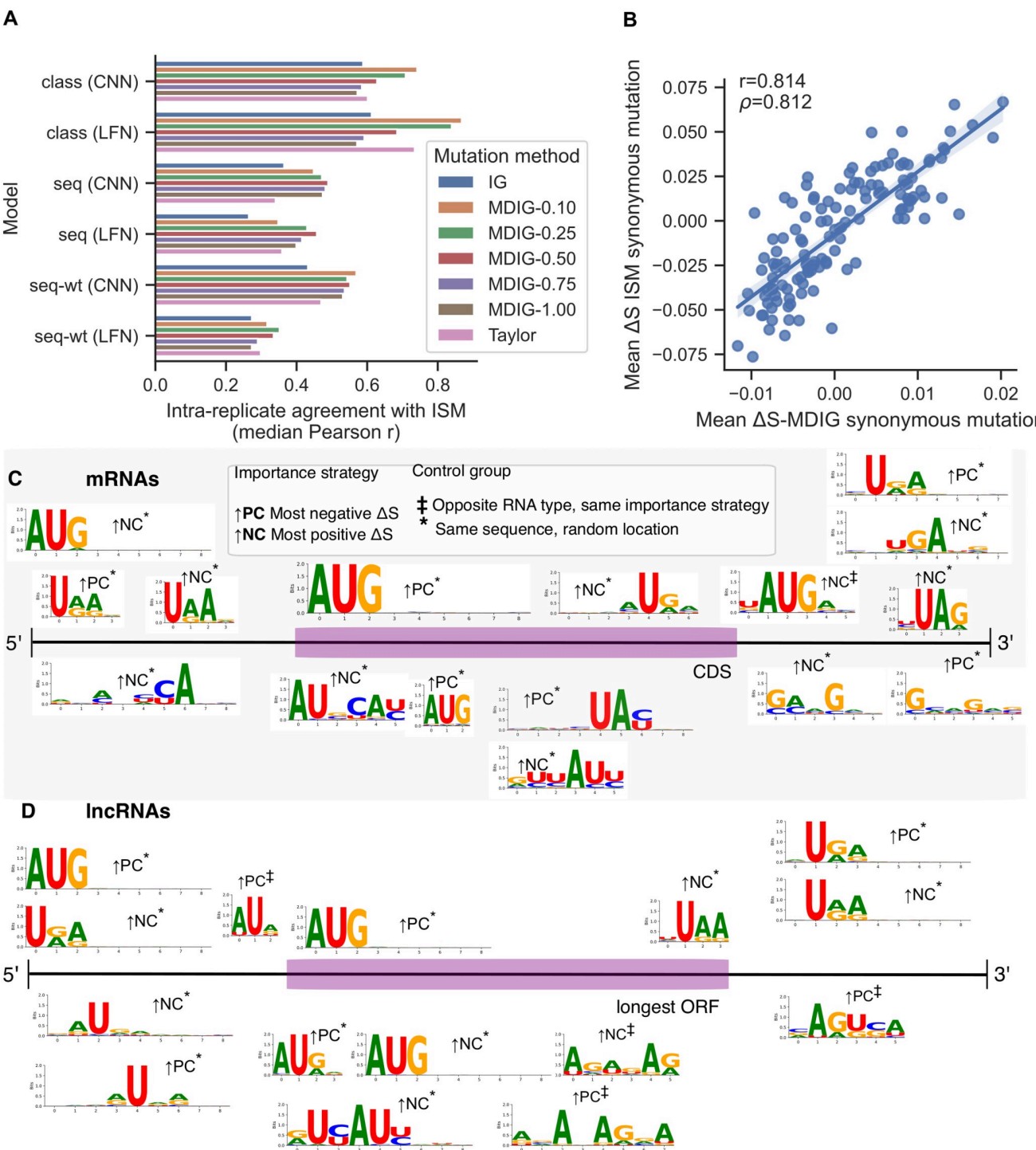

**Fig 6. Gradient-based approximation performance.** (A) Summary results from tuning of $\beta$ hyperparameter for MDIG alongside baseline methods. The intra-replicate agreement according to Pearson correlation of each gradient-based approximation with ISM is summarized using the median across transcripts as a point estimate. (B) Scatter plot of $\Delta S$ for all possible synonymous point mutations, i.e. every wildtype>variant pair differing at one position, from MDIG on the test set (x-axis) versus the same for ISM (y-axis) on the test set. (C) mRNA motifs discovered in our training set with STREME using MDIG importance values from bioseq2seq to determine sequence regions in which to search for enriched signals. Results from unmasked importance are shown above the transcript diagram and those from the masked trials are shown below. (D) lncRNA motifs discovered in the training set using MDIG importance values from bioseq2seq, depicted in the same manner as panel C.

To investigate the implications of MDIG's reduced performance on bioseq2seq-wt, we used the test data and method from the previous section to compute bin-based metagenes from the best MDIG versions and observed that this averaged representation closely captures the same general trends as expected from ISM (Fig I in S1 Text). Across the bioseq2seq replicates, MDIG metagenes have an average correlation of $r = 0.830$ for mRNAs and $r = 0.969$ for lncRNAs with their ISM equivalent, in comparison to $r = 0.997$ and $r = 0.996$, respectively for bioseq2class. For a more detailed evaluation on bioseq2seq-wt we approximated $\Delta S$ for every synonymous point mutation using MDIG on the test set and compared it with the true $\Delta S$ scores from ISM in the form of a scatterplot in Fig 6B. The high correlation between metagenes and codon scores for ISM and MDIG indicates that despite its reduced transcript-level accuracy in predicting bioseq2seq-wt mutation effects, MDIG largely captures the same class-level features as ISM when averaged across examples.

To take advantage of MDIG's improved efficiency relative to ISM for improving the statistical power of motif discovery on our most biologically robust model, we applied MDIG-0.25 on bioseq2seq-wt (LFNet) for the full training set, using only eight integration steps per baseline for increased efficiency. This consists of ∼52k examples, balanced between the two RNA classes. This took slightly more than a GPU-day on NVIDIA V100s, and we estimated that using ISM on this dataset would take 7–22 times as long, depending on the specific hardware used. We used the resulting MDIG mutation effect estimates as a drop-in replacement for ISM importance values in our motif discovery pipeline, with the results presented in Fig 6C for mRNAs and D for lncRNAs. Motifs from masked trials are placed below the transcript diagrams and those from unmasked trials are above. In comparison to the ISM motifs discussed previously, the MDIG motifs better underscore that bioseq2seq-wt places importance on start and stop codons in regions besides the CDS, possibly learning context-specific meaning to these other matches. Start codons are predicted by MDIG to decrease bioseq2seq-wt coding probability in mRNA 5' UTRs while AUGs typically increase coding potential in lncRNA upstream regions. Stop codons typically push the classification towards noncoding in the 5' regions, but significant $\uparrow PC$ scores also occur near certain contextual stop codon motifs in mRNA 5' UTRs. Notably, the UTR motifs are more likely to lack a bias towards a particular frame of the transcript, while ORF features often have a consistent frame bias. This is supportive of the idea that such elements outside the ORF are flagged in part to determine the frame. Some of the masked lncRNA motifs closely resemble those from the unmasked strategy, implying that the masked maxima are nucleotides adjacent to AUGs. This comparative lack of diversity could mean that bioseq2seq-wt largely defines lncRNAs as a class in terms of a lower quality or incorrect context of protein-coding features rather than distinctly 'noncoding' features. It also likely implies that MDIG is most adept at estimating the strongest mutation effects for bioseq2seq-wt, with diminished reliability for less influential signals. On the whole, aggregating over instance-level MDIG scores to drive motif discovery appears to emphasize broad global features on which both MDIG and ISM both place high importance, while revealing additional signals beyond those identifiable with smaller-scale ISM experiments alone. In some cases, the MDIG-driven motifs show statistically significant similarity to known RBP sites [59] as determined by TOMTOM [58], which we note in Fig J in S1 Text As for ISM, the MDIG motifs are shown in greater detail in S4 and S5 Tables.

## Discussion

The genetic code makes it straightforward to predict protein sequences given an mRNA sequence, but our results suggest that requiring a neural network to learn the translation task improves its ability to identify protein-coding RNAs. We hypothesize that translation acts a

regularization strategy by requiring the model to preserve precise positional information in a way that improves its contextual representations. A nearly 3% jump in performance for models trained to explicitly learn the translated sequence (bioseq2seq models), or to implicitly learn it (bioseq2start models), demonstrates the benefit of learning aspects of the genetic code. Our findings are consistent with a related observation from RNAsamba, which performed worse when a network branch processing the longest ORF sequence was ablated [44]. Bioseq2seq differs from RNAsamba in that the translated protein sequence is an output rather than an input to the network. To our knowledge, bioseq2seq is the first attempt to use machine learning to output the encoded protein for an input RNA by explicitly learning the sequence mapping underlying biological translation. It accomplishes this from sequence alone, without introducing prior knowledge about the genetic code. Our models achieve competitiveness with the best prior approaches without engineered sequence features, with bioseq2seq achieving a slightly higher recall and F1 than RNAsamba.

The translation task also appears to significantly improve the quality of the nucleotide-level features identified by our models as predictive of protein-coding potential. The correlation of ISM mutation effects across multiple replicates is considerably higher for bioseq2seq than for bioseq2class. Inter-replicate agreement quantifies the low epistemic uncertainty of mutation effect predictions made by an ensemble of bioseq2seq models. In the absence of experimentally characterized mutation effects, this suggests a robustness in the learned biological rules that can inform the plausibility of insights derived from feature interpretation. Besides this improvement in feature consistency, we found that the translation task confers an additional context-awareness to the model in a way that matches biological intuition. Even though simple features like ORF length are obvious correlates of ribosomal translation activity in the cell, the training process does not automatically impart this mechanistic insight into a neural network. We observed that bioseq2class did not respond strongly towards mutations to either start codons or premature stop codons, suggesting such elements play a minimal role in its classification logic despite its relatively high best-case performance of 0.916. Similarly, although mRNN recognizes start codons, it responded primarily to certain codons found 100–200 nt downstream of the start codon, rather than waiting for the stop codon [45]. Bioseq2seq, however, responds negatively to start codon mutations, stop codon mutations, and nonsense mutations, suggesting that its decision-making is strongly influenced by its learned ORF features. Bioseq2seq's faithful modeling of ORF features and mRNA periodicity improves the chances that it also makes biologically relevant effect predictions with respect to synonymous mutations and motif discovery, which require greater detail. We believe that the translation task steers the network toward more robust and meaningful representations that align with biological knowledge and show relative stability across replicates. In our view, these properties are vital prerequisites to enable a broader reliance on machine learning feature interpretation as a tool for scientific discovery.

Our treatment of gradient-based attributions is a contribution to the ongoing debate in the machine learning literature about the trustworthiness of such methods as neural network explanations. We benchmark gradient-based mutation effect predictions in the biological sequence domain against *in silico mutagenesis*, which is the concrete model response to meaningful sequence perturbations. Strikingly, the translation task appears to adversely affect the quality of gradient approximations, with all such methods achieving relatively poor correlation with ISM for bioseq2seq but acceptable approximation quality in bioseq2class. At a minimum, our results suggest that users of gradient-based feature explanations for genomics should follow a protocol similar to ours to validate gradient-based mutation effect predictions against more expensive but direct input perturbations. It might suggest that for some problems it is better to restrict architecture choices to convolutional neural networks, for which speedups of

ISM exist [60]. More fundamentally, there could be a practical tradeoff between model complexity and accurate gradient approximation such that reduced fidelity of fast model perturbations is a price to pay for the superior classification performance and biologically plausible feature importance values that we observed in bioseq2seq tasks.

We also introduce MDIG as a novel heuristic approximation for ISM, which we demonstrate can improve over Taylor approximation at a constant increase in computational complexity. MDIG is largely based on IG, but uses a more realistic mutation-specific baseline, and only integrates part of the way to the baseline value, staying closer to the original sequence. Despite the limited capacity of MDIG to estimate bioseq2seq mutation effects at the local, i.e. transcript level, we show its utility for identifying the most impactful sequence features at the global, i.e. class-wide level. This is supportive of recent work finding that the usefulness of approximate feature attributions can be improved by ensembling across alternative models [61]. The similarity of important motifs and metagene representations derived from MDIG to their ISM analogues indicates that in aggregate MDIG retains interpretive value even where it does not faithfully model every individual mutation effect. Subject to appropriate validation, MDIG could be used where large-scale ISM experiments are infeasible or as a first-pass method to flag interesting sequences for more detailed review.

Interpreting bioseq2seq using ISM and MDIG revealed putative signals of regulatory information, which emerged purely from the learning process without prior specification. We expect that the sequence features learned to distinguish translated mRNAs from lncRNAs with untranslated ORFs would be informative for promoting ribosomal engagement and would promote translation. We therefore expect that sequence features predicted to increase coding potential will correlate with codon bias. Common methods for assessing codon usage bias, such as Codon Adaptation Index, predict coding sequences according the relative skew of synonymous codons for a particular amino acid towards the codons most common in highly expressed genes. Bioseq2seq learned strong preferences within synonymous groups, as evidenced by consistently high mean value of $\Delta S$ across the entire transcript for specific codons. Codon preferences were noticeably grouped by the nucleotide in the third codon position, with substitutions towards nearly all codons ending in C having a positive mean effect, while nearly all ending in T/U have a negative effect. The existence of codon preference trends along the length of the transcript could reflect the fact that synonymous codon usage can be biased positionally, including towards rare codon clusters [62]. Replacing codons with those preferred by bioseq2seq in the average case could perform a similar function to optimizing based on CAI, but bioseq2seq learns mutation effects in the context of a codon's transcript position and sequence neighborhood. Our mutation effect predictions are therefore a much richer source of information, and future work could test via experiment whether these preferred mutations impact translational efficiency and have potential to guide mRNA sequence optimization. The discovered motifs also reflect sensible biological intuitions, with the MDIG motifs in particular emphasizing AUGs and stop codon trinucleotides in a context-specific way. Some AUGs and stop codon trinucleotides in the 5' UTR increased noncoding potential, consistent with evidence that upstream ORFs act to suppress the translation of the main ORF [24]. Within the 3' UTR, we found that some stop codon trinucleotides promoted coding potential while some decreased it, with the stop codon trinucleotides that diminish coding potential having a bias towards being in frame with the true CDS.

Our demonstration that bioseq2seq can recover potentially translated micropeptides is a proof-of-concept for using machine predictions to explore this cryptic space of the proteome. The identification of micropeptides remains challenging, and although the recovery rate of putative micropeptides from LncPEP is low overall, all of our models outperform RNAsamba on the available data. Crucially, bioseq2seq is not inherently limited to only translating the

longest ORF, which could prove to be a modeling advantage for this application given that many micropeptides are known to be harbored in ORFs other than the longest in a transcript [63]. Increased availability of validated micropeptide annotations and improved procedures for autoregressive decoding—see [64] for an example—could help a future method based on bioseq2seq to achieve higher reliability.

We anticipate that the LFNet architecture will be of broad utility in biological sequence modeling tasks, with frequency-domain multiplication enabling larger context convolutions than in common convolutional architectures and lower computational complexity of $O(N \log N)$ in comparison to transformers. Our extension of GFNet from [65] bridges older signal processing approaches for gene discovery with the flexibility of deep models. Although the LFNet did not systematically outperform dilated CNNs across all tasks, it had better average performance in the best-performing bioseq2seq-wt setting and has benefits for interpretability. LFNet is, to our knowledge, the first demonstration of a useful FFT token mixing approach on biological sequences, and could prove useful in combination with simpler CNN architectures. We also note the complementarity of our method with [66], which, instead of employing the Fourier-transform as a token-mixing method, used it to enforce a smoothness prior for importances on biological sequence models. Other applications of LFNet could include biological sequence data with variable periodic signals, such as nucleosome positioning [67] and gene organization [68], as well as other periodic non-biological data such as music. We designed the LFNet architecture based on an intuition that it could effectively leverage 3-nt periodicity, but such periodic structure is not necessarily an inherent requirement—the GFNet model was originally intended for computer vision.

There are numerous possible follow-up directions based on this work. Future versions could scale to a larger and more phylogenetically diverse dataset beyond the eight mammalian transcriptomes used here, as well as to longer sequence lengths. In this work we have treated coding potential as a binary classification problem, but the methods presented are readily applicable to the more general problem of predicting translational efficiency as a regression problem. The periodicity inductive bias in particular is likely to transfer to this task—Ribo-Seq data is also characterized by a 3-nt periodicity of footprint density, and this has informed the development of many ribosome profiling data analysis tools [69, 70]. The regression setting could also increase the prospects for discovering novel regulatory features, such as in the UTRs, which our model treated as less important than the CDS. A network trained to stratify transcripts according to a quantitative measure of protein expression would likely learn more fine-grained distinctions than one modeling a binary separation between mRNAs and lncRNAs. Finally, our results raise the possibility that general-purpose nucleic acid language models could benefit from joint training with protein foundation models in a similar translation-like setup.

## Methods

### Seq2seq architecture for translation

Our model follows the encoder-decoder sequence-to-sequence (seq2seq) framework common in machine translation of natural languages [49]. We call the model bioseq2seq because it applies the seq2seq paradigm to biological translation—with nucleotides and amino acids rather than human languages as the vocabularies. The output of bioseq2seq is a classification token—⟨PC⟩ for protein coding and ⟨NC⟩ for noncoding—followed by the translated protein in the case of ⟨PC⟩ and nothing in the case of ⟨NC⟩. Note that the network is not provided the location of the CDS, so it must learn to identify valid ORFs and select between potential protein translations.

Training bioseq2seq in this way allows us to test the hypothesis that the translation task will require the model to learn precise representations of each nucleotide, which will in turn help to attribute model decisions to specific sequence patterns. As a comparison with bioseq2seq, we also trained a model for binary classification. This secondary model, which we denote as bioseq2class, has an identical network design to bioseq2seq, but a different training data format, as it was trained to output only the classification token without the additional protein product for mRNAs. Although including a decoder is somewhat atypical when producing a single output classification, we do this to enable a direct comparison between the training tasks under a common architecture. The role of the decoder in the bioseq2class setting is to calculate multi-headed attention distributions over the encoder hidden states, with the pre-pended 'start-of-sentence' token playing a similar role to the '[CLS]' in encoder-only classification setups. We developed our models in PyTorch based on a fork of the OpenNMT-Py repository for machine translation [71].

We also experimented with re-weighting the vanilla machine translation loss function to prioritize accurate prediction of earlier characters in the bioseq2seq decoding, particularly the classification token. For the bioseq2seq-wt objective, the per-position cross entropy loss for each transcript is weighted by a truncated discrete exponential distribution with probability mass function

$$p(i, \lambda) = \frac{(1 - e^{-\lambda})e^{-\lambda i}}{1 - e^{-\lambda N}} \quad \forall i = 0 \ldots N - 1$$

where $i$ is the output position, $N$ is the protein length and $\lambda$ controls the decay rate with position. The overall loss is then computed as the affine combination $\sum_{i=0}^{N-1} \ell(x_i, \hat{x}_i)p(i, \lambda)$ where $\ell(x_i, \hat{x}_i)$ is the cross-entropy loss between predicted character $\hat{x}_i$ and ground truth character $x_i$. This choice of weighting is based on an intuition that each successive amino acid character will contain less useful supervisory information for protein coding potential. A short prefix of amino acids should be sufficient to disambiguate which ORF is being translated, and beyond a certain point the protein generation task may compete with the classification task. Importantly, as $\lambda \to 0$ the weights approach a uniform distribution over positions and as $\lambda \to \infty$ the objective places all weight on the first character. In this sense, $\lambda$ interpolates between the bioseq2seq objective in which all positions contribute equally to the loss, and the bioseq2class objective in which only the leading classification token matters.

## Local Filter Network

We initially experimented with transformer neural networks [49] for both the encoders and decoders but failed to produce competitive models, as biological sequences incur excessive memory costs as model sizes and sequence lengths grow. In these experiments, we found that the transformer encoders for bioseq2seq learned self-attention heads which principally attended to a small number of relative positional offsets while calculating the input embeddings. Additionally, feature attributions showed evidence of a strong 3-nucleotide periodicity (See Fig K in S1 Text).

A variety of recent papers have introduced efficient architectures which aim to preserve the ability of transformers to globally mix information at lower computational cost. A number of these approaches have used the Fourier transform as a substitute for self-attention, because it is an efficient global operation computable in $O(N \log N)$ time via the fast Fourier transform (FFT) algorithm [72, 73]. One such example for computer vision is the Global Filter Network, which takes the FFT of image patches and applies a learnable frequency-domain filter via elementwise multiplication, before inverting the FFT to return the representation to the time

domain. This operation is equivalent to a global convolution in the time domain by the convolution theorem of the Fourier transform.

As the 3-base periodicity property is localized to coding regions within transcripts, we propose a simple modification to Global Filter Networks by substituting the global FFT with the short-time Fourier transform (STFT). While GFNet operates on non-overlapping patches of the input, we follow common practices for STFT using a stride equal to half the window size and weighting with the Hann function. To emphasize that our modification applies time-frequency analysis to local sequence representations, we refer to this layer as a Local Filter Network (LFNet). A learned weight matrix $W$ is applied equally to each window of the STFT and then the modified frequency content is returned to the time domain via the inverse FFT. A residual term is added to the result to carry along the previous representation. Following [73], we apply the soft-shrink function after the weight multiplication to promote sparsity in the LFNet weights. LFNet layers are only used in the encoder stack of our networks, while the decoder stack consists of transformer decoder layers. This is because PyTorch currently lacks an implementation of causal masking for FFT, as would be necessary to efficiently train an autoregressive model with only LFNet layers. To compare LFNet layers with a more conventional modeling approach, we construct an alternate encoder architecture using CNNs with residual connections, and optionally increase dilation by a factor of two with each successive layer.

## Start codon prediction

We defined a fourth training task called bioseq2start to test the theory that learning the CDS location is the most useful regularization strategy for a model of coding potential. In bioseq2start, the output is an integer corresponding to the index of the start codon within the input sequence mRNA. For lncRNAs, the required output is the last position, where an open reading frame cannot possibly begin. This task differs with bioseq2seq in that it explicitly trains the model to identify the CDS, whereas bioseq2seq implicitly locates the CDS via its protein translation. However, bioseq2start may be a simpler learning task than bioseq2seq, which must also learn the genetic code.

To produce an integer output based the input sequence length, we used a Pointer Network layer [74], which was initially proposed for neural combinatorial optimization problems and later applied to label text spans for question answering [75]. Instead of a stack of transformer decoders, the input encoder representations are passed to a single linear layer which projects each input encoder representation down to a scalar. A softmax function over the input positions transforms these values to a probability distribution and the loss is computed as a cross-entropy between this predicted distribution and the true start codon position. For predicting coding potential from bioseq2start output, we considered a prediction to be $\langle NC \rangle$ if its argmax was the last sequence position and $\langle PC \rangle$ otherwise. Where a full $P(\langle PC \rangle)$ is desired, it can be computed as the sum of the probabilities for all positions before the last.

## Dataset

We built training and evaluation data sets using available RefSeq transcript and protein sequences for eight mammalian species: human, gorilla, rhesus macaque, chimpanzee, orangutan, cattle, mouse, and rat from RefSeq release 200 [76]. We collected all RNA sequences annotated as mRNA or lncRNA and excluded transcripts over 1200 nucleotides (nt) in, which reduces the available data to 63,272 transcripts. Next, we linked each mRNA with the protein translation identified by RefSeq and partitioned the data into 80/10/10 training/validation/testing splits. To maximize the diversity of the dataset, we included transcripts with predicted

coding status (XR and XM prefixes in RefSeq), as well as the curated transcripts (NM and NR). For the training set, we used a balanced split between mRNAs and lncRNAs, selecting the split to equalize the length distribution of the two classes as much as possible. We ran CD-HIT-EST-2D to exclude from the test and validation sets all transcripts that exceed 80% similarity with any transcript in the training set [77]. Because CD-HIT-EST-2D uses greedy clustering, banded alignment, and kmer-based heuristics to avoid exhaustive pairwise alignments, we further refine the test set by using an all-by-all BLAST search between the testing set and both the training and validation sets followed by Needleman-Wunsch alignment to identify homology above the 80% missed by CD-HIT. Finally, twenty coding transcripts with incomplete start codon annotations were excluded to avoid disadvantaging bioseq2start. The resulting test set contains 1913 lncRNAs and 1791 mRNAs. For certain computationally expensive analyses we subsampled all curated noncoding transcripts and an equal amount of curated coding transcripts, then applied the extra level of filtering to the test set only. These are referred to as the "verified" test and validation sets above.

## Hyperparameter tuning and training

We used dynamic batch sizes, so that RNA-protein training pairs were binned based on approximate length to reduce the amount of padding. The maximum number of input tokens per batch was set to 9000 for both model types, and eight steps of gradient accumulation was used to increase the effective batch size. All models were trained to minimize a log cross-entropy objective function computed from each amino acid character in the output.

The hyperparameters including number of encoder and decoder layers, model embedding dimension, learning rate schedule, L1 sparsity parameter for the Softshrink activation in LFNet, and dilation factor for CNNs, were tuned via the Bayesian Optimization Hyperband (BOHB) algorithm provided in the Ray Tune library [50, 78]. Candidate models were trained in parallel on four GPUs with one GPU per model. To enable a fair comparison between the training objectives and architecture types, hyperparameter tuning was run separately for each combination thereof. We then produced five replicates for each of bioseq2seq, bioseq2seq-wt, bioseq2class, and bioseq2start using both LFNet and CNN models. For further details on hyperparameter tuning and model training see the S1 Text.

## Mutation effect prediction

Estimating the effects of sequence mutations can provide insight into the importance that the model assigns each input nucleotide. The gold standard for computationally scoring mutation effects, known as *in silico mutagenesis* (ISM), requires comparing the model predictions for all single-nucleotide variants with that of the original sequence [54]. The computational expense of this procedure—$3L$ model evaluations for a transcript of length $L$—motivates us to explore the effectiveness of gradient-based approximations.

Below we refer to the network output function by $S$, and the output gradient with respect to its input as $\nabla_x S(x)$. In general, $S$ can be any scalar output, and here we use $S = l_{\langle PC \rangle} - l_{\langle NC \rangle}$, the difference in logits, i.e unnormalized log probabilities, for the RNA classification tokens in the first decoding position. We denote the two sequences being compared as $x, x' \in \mathbb{R}^{L \times V}$ for one-hot encodings of categorical variables and $V$ as the input vocabulary size.

**Taylor series approximation.** The simplest ISM surrogate begins with a Taylor expansion of a differentiable function $F$ around a point of interest $x'$.

$$F(x') \approx F(x) + \nabla_x F(x)^\top (x' - x) + o(\|x' - x\|)$$

In this fashion, we can expand around $S$ and discard all higher order terms for a first-order Taylor approximation of the difference in $S$.

$$\Delta S(x', x) = S(x') - S(x) \approx \nabla_x S(x)^\top (x' - x) \tag{1}$$

Since we confine our analysis to single-mutations, this simplifies to

$$\Delta S(x\{i, j \rightarrow b\}, x) \approx \frac{\partial S(x)}{\partial x_{ib}} - \frac{\partial S(x)}{\partial x_{ij}} \tag{2}$$

where $x\{i, j \rightarrow b\}$ is the result of mutating RNA $x$ at position $i$ from nucleotide $j$ to $b$. Thus, all $3L$ values are computable from $\nabla_x S(x)$ in just one forward/backward pass of the network.

**Integrated Gradients.** The input gradient represents only an infinitesimal change in the input-output behavior of the network, rather than the effect of a full character substitution as in ISM. When a local approximation does not accurately describe the global function behavior, this is a well known limitation called gradient saturation [79]. As a more sophisticated proxy for ISM, we adapt a procedure called Integrated Gradients (IG), which was designed to reduce the the effect of gradient saturation and satisfies several desirable axioms for importance metrics [80]. IG uses a baseline input $x'$ and computes an integral using input gradients for a differentiable function $F$ along the linear path between $x'$ and $x$.

$$IG(x, x')_{ik} = (x_{ik} - x'_{ik}) \int_{\alpha=0}^{1} \frac{\partial F(x' + \alpha \times (x - x'))}{\partial x_{ik}} \, d\alpha \tag{3}$$

Common choices of $x'$ are an all-zero baseline and a uniform probability mass function [0.25, 0.25, 0.25, 0.25] over the four bases in all input positions. Importantly, Integrated Gradients satisfies the equality $\sum_i \sum_k IG(x, x')_{ik} = F(x) - F(x')$, known as the completeness axiom. We can define a rudimentary mutation effect approximation by subtracting the IG importance scores for the endogeneous base $j$ and a mutation $b$ as follows,

$$\Delta S(x\{i, j \rightarrow b\}, x) \approx IG(\mathcal{U}(x), x)_{ib} - IG(\mathcal{U}(x), x)_{ij} \tag{4}$$

where $\mathcal{U}(x)$ represents the uniform baseline. Note the order of arguments, which re-frames the baseline as the destination rather than the source.

**Mutation-Directed Integrated Gradients (MDIG).** Eq 4 represents a naive attempt to derive mutation scores from IG attribution. However, any such interpretation suffers from two glaring limitations (1) The uniform baseline is non-biological (2) IG completeness only guarantees that the attributions *sum* to the function differences of $x$ and $x'$ without any stronger guarantees on each feature attribution $IG_{ik}$. The latter point speaks to the difficulty of repurposing an interpretability method centered around an arbitrarily distant baseline $x'$ into an approximation method for $3L$ single-nucleotide perturbations. In other words, the IG scores corresponding to a large perturbation *are not decomposable* into the equivalent scores for multiple smaller perturbations.

To improve the correspondence between IG and ISM, we first note a relationship between IG and Taylor-approximation. Given one-hot encodings for $x$ and $x'$ with $x'$ differing from $x$ in one nucleotide position $i$, we know that $(x_{ik} - x'_{ik}) = 1$ when $k = j$ where $j$ is the endogeneous character, $(x_{ik} - x'_{ik}) = -1$ when $k = b$, the mutated character, and $(x_{ik} - x'_{ik}) = 0$ otherwise. The column-wise sum $\sum_k IG(x, x')_{ik}$ is therefore equivalent to a dot-product between a vector of a particular form, (e.g. $\begin{bmatrix} 1 & 0 & -1 & 0 \end{bmatrix}$ with $x$ having A at position $i$ and $x'$ having G) and a row of the integral from IG in Eq 3. This is similar to the Taylor approximation in Eq

2, in which the partial derivative (rather than some related integral) for the endogeneous character is subtracted from that of the mutant character.

Based on this loose analogy, we begin with IG baselines consisting of of all nucleotide $b$ of the same length as $x$, e.g. a sequence of all guanines, referring to this as $poly(b)$. Then, we introduce a hyperparameter $\beta \in (0, 1]$ for a final baseline of

$$X(b, \beta) = \beta \cdot poly(b) + (1 - \beta) \cdot x \tag{5}$$

with $\beta$ meant to balance an inherent tradeoff in the integration path. On the one hand, we must remain close enough to the original sequence for accumulated gradients to carry meaningful information about small, localized perturbations to the original sequence rather than the distant $poly(b)$. On the other, we must advance far enough towards $poly(b)$ so that each $x' - x$ approaches vectors like $\begin{bmatrix} 1 & 0 & -1 & 0 \end{bmatrix}$ and the desired interpretation of a full point-mutation. As the point best balancing this tradeoff is unclear from first principles, we tune $\beta$ on a validation set to maximize the correlation with ISM.

$$\Delta S(x\{i, j \rightarrow b\}, x) \approx IG(X(b, \beta), x)_{ib} + IG(X(b, \beta), x)_{ij} \quad \forall b \in \{A, C, G, U\} \tag{6}$$

Note that this requires an IG evaluation for each $poly(b)$ baseline rather than a single baseline as in Eq 4. We dub the resulting method Mutation Directed Integrated Gradients (MDIG).

## Evaluation metrics

**Classification.**　The Matthews Correlation coefficient is defined as

$$MCC = \frac{TP \times TN + FP \times FN}{\sqrt{(TP + FP)(TP + FN)(TN + FP)(TN + FN)}}$$

where $TP$, $TN$, $FP$ and $FN$ are respectively, the number of true positives, true negatives, false positives, and false negatives. MCC is used for datasets with a significant class imbalance and may be a preferable metric over F1, which is more commonly used in such situations [81].

**Gradient attributions.**　We compared $L \times 3$ vectors (sequence length $\times$ 3 possible mutations) of mutation effect predictions using the metrics

$$\text{Pearson}(x, y) = \frac{cov(vec(x), vec(y))}{\sigma(vec(x))\sigma(vec(y))}$$

$$\text{Median position-wise cosine similarity}(x, y) = median([cos(x_1, y_1), cos(x_2, y_2), \cdots cos(x_L, y_L)])$$

Where $cov()$ is the covariance, $\sigma()$ is the standard deviation, $vec()$ is the vectorization operator, which flattens a matrix into a vector, $cos(x, y) = \frac{x \cdot y}{||x||||y||}$, and $x_i$ refers to a row vector of matrix $x$. The inter-replicate agreement is

$$\text{Inter-replicate agreement}(x) = |\binom{S}{2}|^{-1} \sum_{i,j \in \binom{S}{2}} metric(mut(x)_i, mut(x)_j)$$

where $\binom{S}{2}$ is the set of all possible subsets of cardinality 2 from the set of model replicates $S$, $mut(x)_i$ is the mutation effect prediction coming from replicate $i$ for a given RNA $x$, and $metric$ is one of Pearson r or median position-wise cosine similarity, as described above. The

agreement with ISM is defined with intra-replicate comparisons.

$$\text{Agreement with } \text{ISM}(x) = |S|^{-1} \sum_{i \in S} metric(ISM(x)_i, mut(x)_i) \tag{14}$$

### Motif discovery from mutation effect predictions

To uncover sequence elements salient to bioseq2seq predictions, we converted ISM scores into importance scores for the endogenous characters. In particular, we set the importance score of an endogenous base with respect to a given class as equal to the absolute value of $\Delta S$ for the strongest mutation in the direction of the *counterfactual* class, following [82] which used the equivalent from regression models for visualizing importance. For example, an endogenous $x_i$ within an mRNA was defined as contributing towards a true positive classification of $\langle PC \rangle$ to the extent that substituting any of the three alternate bases in position $i$ produces a highly negative $\Delta S$, which pushes the prediction towards a false negative of $\langle NC \rangle$. We calculated importance using both classes on all transcripts. For instance, we looked for strong local contributions towards a prediction of $\langle PC \rangle$ within annotated lncRNAs.

For a given importance setting, we then extracted a window of 10 nt upstream and 10 nt downstream around the position with the highest importance score for a total length of 21 nt. This process was run separately for mRNA 5' and 3' UTRs and CDS sequences, and similarly for lncRNAs using the longest ORF and its upstream and downstream regions. We used the STREME motif discovery tool to efficiently identify sequence motifs occurring frequently in these regions of interest [57]. STREME estimates p-values for motifs, and after collecting all discovered sequence logos, we reported all that were significant at the 0.001 level after applying the Bonferroni correction for multiple testing.

## Supporting information

**S1 Text. Supplementary methods.** Additional methodological details and results for hyperparameter tuning, training, and interpretation of models. Includes supplementary tables and figures as referenced throughout.
(PDF)

**S1 Table. LncPEP annotation.** Transcript names from LncPEP database are mapped to NCBI accessions and micropeptide sequences.
(XLSX)

**S2 Table. Motifs derived from unmasked in silico mutagenesis importance on the testing set.**
(PDF)

**S3 Table. Motifs derived from masked in silico mutagenesis importance on the testing set.**
(PDF)

**S4 Table. Motifs derived from unmasked MDIG importance on the training set.**
(PDF)

**S5 Table. Motifs derived from masked MDIG importance on the training set.**
(PDF)

## Author Contributions

**Conceptualization:** Joseph D. Valencia, David A. Hendrix.

**Data curation:** Joseph D. Valencia.

**Formal analysis:** Joseph D. Valencia, David A. Hendrix.

**Funding acquisition:** David A. Hendrix.

**Investigation:** Joseph D. Valencia.

**Methodology:** Joseph D. Valencia.

**Resources:** David A. Hendrix.

**Software:** Joseph D. Valencia.

**Supervision:** David A. Hendrix.

**Validation:** Joseph D. Valencia.

**Visualization:** Joseph D. Valencia.

**Writing – original draft:** Joseph D. Valencia.

**Writing – review & editing:** Joseph D. Valencia, David A. Hendrix.

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
