## [Decision Letter · Decision Letter 0]

6 Jun 2023

Dear Dr. Hendrix,

Thank you very much for submitting your manuscript "Improving deep models of protein-coding potential with a Fourier-transform architecture and machine translation task" for consideration at PLOS Computational Biology.

As with all papers reviewed by the journal, your manuscript was reviewed by members of the editorial board and by several independent reviewers. In light of the reviews (below this email), we would like to invite the resubmission of a significantly-revised version that takes into account the reviewers' comments.

We cannot make any decision about publication until we have seen the revised manuscript and your response to the reviewers' comments. Your revised manuscript is also likely to be sent to reviewers for further evaluation.

Sincerely,

Zhaolei Zhang

Academic Editor

PLOS Computational Biology

Jian Ma

Section Editor

PLOS Computational Biology

Reviewer's Responses to Questions

**Comments to the Authors:**

Reviewer #1: In this work, the authors augment a model to predict protein-coding potential with an additional task to output the sequence of the protein produced in case of protein-coding genes. The authors interpret the models using ISM and MDIG, a variant of Integrated Gradients developed by the authors to approximate ISM. They show that augmenting the model with the sequence translation task improves model interpretability. Overall, the paper is well written and experiments are convincing. I have some comments regarding the methods and suggestions for experiments.

Major:

- In my opinion, a more direct solution would be to predict only the start and end indices of the protein-coding transcript. That would provide the model the same information as predicting the protein sequence, as the mapping from (start,end) to sequence is deterministic. This would simplify to predicting the start and end indices auto-regressively, similar to methods employed for extracting subsequences for in the case of question-answering (see https://arxiv.org/pdf/1608.07905.pdf). Could the authors comment on their design choice? If possible, can the authors show that outputting the protein sequence performs better compared to simply outputting the start and end coordinates of the CDS.

- Can the authors also report metrics such as AUROC/AUPRC that would not depend on the specific threshold used for binary classification.

- Do the authors treat the relative weights of the two tasks (binary + sequence generation) as a hyper parameter? If not, why?

- The authors mention that "when bioseq2seq was allowed to predict a full-length protein rather than...the classification performance of the best model deteriorated slightly...": I am not sure why this is the case. My understanding is that the binary prediction is separate from the protein prediction (indeed it is modelled as such), and the predicted protein sequence is conditioned on the case that the prediction is protein-coding. The accuracy of the binary task should not be affected by the sequence generation.

- Many of the questions and hypotheses posed in the discussion can be explicitly tested using in-silico experiments:

= the authors can test how the prediction changes as the length of the CDS is reduced by changing the position of the start and/or stop codons

= the authors can test how the presence/absence of upstream ORFs (their length, relative position) is predicted to affect coding potential

- The authors may choose to attempt to answer some other interesting questions using in-silico experiments: in how many edits can a coding transcript be converted to non-coding (without breaking start codons or creating stop codons). The converse (lncRNA->mRNA) conversion can also be tried.

Minor:

- Specify full form of MCC in Table 1

- Table 2 is not cited in the main text

- typo "as an transcript-level" (Page 10)

- The masking strategy for interpretation bears some resemblance to https://pubmed.ncbi.nlm.nih.gov/35966405/, and the authors may choose to explore the interpretation strategies outlined in the paper

- more potential explanation on why UGG may increase non-coding potential would be useful

Reviewer #2: In this manuscript, the authors proposed bioseq2seq, a seq2seq formulation for predicting protein-coding potential with deep neural networks and demonstrate that simultaneously learning translation from RNA to protein improves classification performance relative to a classification-only training objective. Bioseq2seq introduces Local Filter Network (LFNet), a computationally efficient network layer based on the short-time Fourier transform. LFNet is a network architecture with an inductive bias for modeling the three-nucleotide periodicity apparent in coding sequences. the authors incorporate LFNet within an encoder-decoder framework to test whether the translation task improves the classification of transcripts and the interpretation of their sequence features. The resulting model is then used to compute nucleotide-resolution importance scores, revealing sequence patterns that could assist the cellular machinery in distinguishing mRNAs and lncRNAs.

See the detailed comments below:

1. The output of bioseq2seq is either <pc> for protein coding and <nc> for noncoding followed by the translated protein in the case of <pc> and nothing in the case of <nc>. However, bioseq2seq’s formulation seems unnecessary. In principle, if the status (i.e., <pc> or <nc>) is predicted, the sequence inside (i.e., the translated protein or nothing) is deterministic. That said, simple binary classification seems sufficient and seq2seq is overkill.

2. According to Fig 2, the predicted proteins deviate from the true protein in the status <pc>, suggesting that the seq2seq model is not as reliable as translating the protein using the deterministic rule.

3. The classification performance in Table 1 is not very convincing because the baseline methods are not comprehensive and relatively weak. That said, the major claim of the paper “simultaneously learning translation from RNA to protein improves classification performance relative to a classification-only training objective” cannot be well supported. More comprehensive evaluation is needed.

4. The core feature of LFNet is its learned frequency-domain filters whose frequency bin equivalent to a period of 3 nt. Though it sounds novel, a trivial alternative can be using CNN with filter kernel size=3 and stride=3. Justification is needed why the proposed frequency-domain filters are better. Maybe an ablation study is needed</pc></nc></pc></nc></pc></nc></pc>

Reviewer #3: In this manuscript, the authors developed a new approach for protein-coding potential prediction from RNA sequences using a FFT-based frequency domain neural network architecture and a machine translation task. Using FFT on this task is a novel idea that utilizes the 3nt periodicity of the protein-coding sequence. The models proposed by the authors achieve comparable performance with prior art such as RNAsamba without relying on pre-selected input features. In addition, the authors performed extensive analysis to interpret the model, including testing a new approach of using an integrated gradient method to approximate the predicted mutation effect from saturated in silico mutagenesis (ISM). Overall, this work represents a meaningful advance in the field, and the analyses were overall carefully performed. My detailed comments are below:

Major points:

1. In this study, the authors collected sequences from eight mammalian species, which were divided into training, validation, and test sets. Based on my understanding of the method section, even though the authors made an effort to reduce the similarity between training and test set by excluding those with >80% identity, the test set may still contain sequences with strong homology to the training set sequences. Can the authors make sure that the test set sequences do not liftover to training set sequences? Can the test set be stratified by maximal similarity to the training set and compare performance across different levels of similarities?

2. For translation tasks, what happens when the produced translation does not exactly match the ground truth? It seems like a task that should achieve 100% accuracy if the translation start site is correct.

Minor points:

1. The Fourier transform has a direct relationship to convolution, as convolution can be implemented with the FFT transform. It would be worthwhile to discuss the differences and advantages of this particular approach compared to the traditional convolution approach, considering this connection.

2. How much performance improvement does Softshrink in the authors' architecture provide, which is an unusual and interesting design, compared to using a regular activation function after IFFT?

3. Does the inductive bias favor the utilization of protein-coding regions over UTRs? Is it as effective as convolution-based approaches in detecting translation start signals? The model appears to capture the downstream 2bp of the Kozak sequence. What about the Kozak sequence upstream of ATG?

4. Can the performance of the pretrained model RNAsamba be compare to the model proposed by authors?

**Have the authors made all data and (if applicable) computational code underlying the findings in their manuscript fully available?**

Reviewer #1: Yes

Reviewer #2: Yes

Reviewer #3: Yes

PLOS authors have the option to publish the peer review history of their article (what does this mean?). If published, this will include your full peer review and any attached files.

Reviewer #1: No

Reviewer #2: No

Reviewer #3: No
---

## [Decision Letter · Decision Letter 1]

18 Sep 2023

Dear Dr. Hendrix,

We are pleased to inform you that your manuscript 'Improving deep models of protein-coding potential with a Fourier-transform architecture and machine translation task' has been provisionally accepted for publication in PLOS Computational Biology.

Best regards,

Zhaolei Zhang

Section Editor

PLOS Computational Biology

Jian Ma

Section Editor

PLOS Computational Biology

Reviewer's Responses to Questions

**Comments to the Authors:**

Reviewer #1: The authors have satisfactorily addressed all of my comments.

Reviewer #3: Thank you for the revision which has addressed all my previous points.

**Have the authors made all data and (if applicable) computational code underlying the findings in their manuscript fully available?**

Reviewer #1: Yes

Reviewer #3: None

PLOS authors have the option to publish the peer review history of their article (what does this mean?). If published, this will include your full peer review and any attached files.

Reviewer #1: No

Reviewer #3: No

---

## [Editor Report · Acceptance letter]

5 Oct 2023

PCOMPBIOL-D-23-00630R1 

Improving deep models of protein-coding potential with a Fourier-transform architecture and machine translation task

Dear Dr Hendrix,

I am pleased to inform you that your manuscript has been formally accepted for publication in PLOS Computational Biology. Your manuscript is now with our production department and you will be notified of the publication date in due course.

With kind regards,

Anita Estes
